# Nitisinone Treatment Affects Biomarkers of Bone and Cartilage Remodelling in Alkaptonuria Patients

**DOI:** 10.3390/ijms241310996

**Published:** 2023-07-01

**Authors:** Federica Genovese, Peder Frederiksen, Anne-Christine Bay-Jensen, Morten A. Karsdal, Anna M. Milan, Birgitta Olsson, Mattias Rudebeck, James A. Gallagher, Lakshminarayan R. Ranganath

**Affiliations:** 1Nordic Bioscience A/S, 2730 Herlev, Denmark; pef@nordicbio.com (P.F.); acbj@nordicbio.com (A.-C.B.-J.); mk@nordicbio.com (M.A.K.); 2Department of Clinical Biochemistry and Metabolic Medicine, Liverpool Clinical Laboratories, Liverpool University Hospitals NHS Foundation Trust, Liverpool L69 3BX, UK; anna.milan@liverpoolft.nhs.uk (A.M.M.); lrang@liverpool.ac.uk (L.R.R.); 3Garriguella AB, 179 62 Ekerö, Sweden; birgitta.spellerberg@gmail.com; 4OnPoint Science AB, 118 49 Stockholm, Sweden; mattias.rudebeck@immedica.com; 5Institute of Ageing and Chronic Disease, University of Liverpool, Liverpool L69 3BX, UK; j.a.gallagher@liverpool.ac.uk

**Keywords:** alkaptonuria, biomarkers, collagen

## Abstract

Nitisinone has been approved for treatment of alkaptonuria (AKU). Non-invasive biomarkers of joint tissue remodelling could aid in understanding the molecular changes in AKU pathogenesis and how these can be affected by treatment. Serological and urinary biomarkers of type I collagen and II collagen in AKU were investigated in patients enrolled in the randomized SONIA 2 (NCT01916382) clinical study at baseline and yearly until the end of the study (Year 4). The trajectories of the biomarkers over time were observed. After treatment with nitisinone, the biomarkers of type I collagen remodelling increased at Year 1 (19% and 40% increase in CTX-I and PRO-C1, respectively), which was potentially reflected in the higher degree of mobility seen following treatment. The biomarkers of type II collagen remodelling decreased over time in the nitisinone group: C2M showed a 9.7% decline at Year 1, and levels then remained stable over the following visits; CTX-II showed a 26% decline at Year 3 and 4 in the nitisinone-treated patients. Nitisinone treatment induced changes in biomarkers of bone and cartilage remodelling. These biomarkers can aid patient management and deepen our knowledge of the molecular mechanisms of this rare disease.

## 1. Introduction

Alkaptonuria (AKU), also known as black bone disease due to its manifestation of hyperpigmentation, is a rare autosomal recessive disorder caused by the deficiency of homogentisate 1,2-dioxygenase (HGD), which causes an accumulation of homogentisic acid (HGA), some of which is deposited as ochronotic pigment in connective tissues, particularly weight-bearing cartilages [1,2]. Ochronotic pigment deposition in cartilage alters the biomechanical [3] and biochemical properties [4] of cartilage, leading to resorption of the subchondral bone plate and calcified cartilage, possibly due to stress shielding. Although the clinical phenotype is predominantly premature severe spondyloarthropathy, other clinical manifestations include osteopenia [5], rupture of tendons, muscles and ligaments, hearing loss, renal failure, and aortic valve disease, which together cause severe disability [6]. AKU presents in approximately 1 in every 250,000 to 1 million people worldwide [7], and until recently, there was no disease-modifying therapy.

After the clinical studies, Suitability of Nitisinone in Alkaptonuria 1 (SONIA 1) [8] and 2 (SONIA 2) [9], in which efficacy and safety of once-daily nitisinone for patients with AKU were studied by the DevelopAKUre consortium, it was demonstrated that 10 mg nitisinone daily reduced the accumulation of HGA by 99.7% and improved clinical symptoms [9]. This resulted in nitisinone being approved by the European Medicine Agency as the first disease-modifying treatment for adults with AKU.

In AKU mice, nitisinone is completely effective in preventing ochronosis if administered from birth [10] and arrests further ochronosis if administered from mid-life [11]. The results of SONIA 2 have demonstrated that the administration of nitisinone in patients with clinical manifestations of AKU can slow disease progression (slower increase in the AKU Severity Score Index or AKUSSI) [9].

Non-invasive biomarkers of connective tissue damage in AKU would be helpful in detecting the earliest sign of the disease taking hold. The SOFIA (Subclinical Ochronosis Features in Alkaptonuria) study concluded that ochronosis can be present before 20 years of age [12]. It is, however, still not known when the disease process ochronosis first appears in AKU as included patients were at least 16 years old. Establishing when ochronosis and biomarkers are increased in childhood would allow an understanding regarding the possible use of nitisinone early in life. Furthermore, since nitisinone is an inhibitor of 4-hydroxyphenylpyruvate dioxygenase, which halts the production of HGA and decreases ochronosis, biomarkers of connective tissue remodelling could describe the downstream effect of the dampening of ochronosis pigment accumulation on the bone and cartilage turnover.

Non-invasive biomarkers of extracellular matrix (ECM) remodelling were previously tested in serum and urine of 40 patients enrolled in the 4-week international, multicentre, randomised, open-label, no-treatment controlled, parallel-group, dose-response SONIA 1 study [13]. Among the markers measured in that study, four biomarkers of type I and II collagen turnover (CTX-I and PRO-C1, reflecting bone resorption and bone formation, respectively, and CTX-II/Creat and C2M, reflecting cartilage remodelling) were present at different levels in serum and urine of patients with AKU compared to controls [13]. The same biomarkers were also evaluated in SOFIA [12].

In the current study, the above biomarkers were tested in the larger population of SONIA 2 to observe whether the treatment with nitisinone altered the dynamics of cartilage and bone remodelling and whether the changes were related to the severity of the disease.

## 2. Results

### 2.1. Demographic and Clinical Parameters

In total, 138 patients with AKU were enrolled in the 4-year, open-label, evaluator-blind, randomised, no-treatment controlled, parallel-group SONIA 2 study. Of these, 69 were treated with 10 mg of nitisinone daily for 4 years, except for 10 patients who developed corneal keratopathy and received 2 mg daily instead [9] and 69 who were not treated. Since the treatment causes a change of colour in the urine, the patients could not be masked, hence the absence of a placebo-controlled group.

The majority of the SONIA 2 population was Caucasian, with 60% being males. The median age was 50 years, and the median BMI was 26 kg/m^2^ at baseline. No demographic and clinical parameters differed significantly between the control and treated groups at baseline (Table 1).

### 2.2. Effect of Nitisinone Treatment on the ECM Biomarkers

None of the studied biomarkers showed a significant difference after nitisinone treatment (end of study), but CTX-II/Creat levels increased significantly in the urine of controls and not that of treated patients (Table 2).

However, analysing the trajectories over time, levels of the biomarkers of bone remodelling (CTX-I and PRO-C1) increased significantly in patients treated with nitisinone compared to controls after one year of treatment (19% (95% CI [5%, 36%]) and 40% (95% CI [19%, 65%]), respectively), returning to levels like the control group at later time points (Figure 1). The same trajectory was observed for ALP (Appendix A). On the contrary, levels of the biomarkers of cartilage remodelling (CTX-II/Creat and C2M) decreased in patients treated with nitisinone compared to controls. Specifically, CTX-II/Creat in urine increased in both groups up to the 2-year visit, after which the levels decreased in the nitisinone group, which had levels of CTX-II/Creat suppressed by 27% (95% CI [11%, 40%]) compared to controls at the 3-year visit and by 26% (95% CI [8%, 41%]) at the Year 4 visit. The trajectories of the biomarker of cartilage remodelling C2M diverged at early time points, and in patients treated with nitisinone the levels of C2M were suppressed by 9.4% (95% CI [3%, 15%]) at year 1 (Figure 1).

It was then investigated whether the trajectories of the ECM biomarkers and ALP were different in patients stratified by sex, age (≤55 or >55 years), cAKUSSI (above or below the median), and concomitant antiresorptive treatment at baseline (Appendix A).

When stratifying by age, the lowering of CTX-II/Creat at Year 1, 3, and 4 in the nitisinone-treated patients was significant only in the younger patients. Further, the lowering in CTX-II/Creat at Year 4 was only significant in males. When looking at C2M, the difference in the levels of the biomarker at Year 4 in the nitisinone-treated group was significant only in males and in patients with cAKUSSI below the median, i.e., the less severely ill patients. There was no significant interaction of the concomitant use of antiresorptive treatment for any of the biomarkers, and there was no significant interaction between treatment and age, sex, or cAKUSSI at baseline for the biomarkers of type I collagen turnover.

### 2.3. Levels of ECM Biomarkers Correlated to Demographic and Clinical Parameters

Levels of the ECM biomarkers at baseline in patients stratified by sex, age, cAKUSSI, and antiresorptive treatment (bisphosphonates, vitamin D, systemic glucocorticoids, and calcium) are shown in Table 3, Table 4 and Table 5 and Appendix A. Levels of ALP at baseline in patients stratified by sex, age, cAKUSSI, and antiresorptive treatment are shown in Appendix A.

Levels of CTX-I were elevated in males, and levels of CTX-II/Creat were elevated in older patients and in patients with higher cAKUSSI. The cAKUSSI was also higher in older patients (Table 4).

When looking at baseline correlations of the biomarkers of bone and cartilage remodelling with the components of the cAKUSSI related to bone and joint health (osteoarticular disease in joints and spine, pain in joints and spine, number of experienced adult fractures, and osteopenia of hips [bone density in g/cm^2^ and its T-score]), CTX-I and C2M were not significantly correlated with any of the parameters, and PRO-C1 had a weak negative correlation with the bone density T-score (Spearman r = −0.24, *p* = 0.008), while CTX-II/Creat correlated moderately with all parameters except the number of fractures (Spearman r = 0.29–0.51, *p* = 0.02–*p* < 0.0001) and had a moderate negative correlation with the bone density T-score (Spearman r= −0.5, *p* < 0.0001). PRO-C1 correlated with both CTX-I (Spearman r = 0.72, *p* < 0.0001) and CTX-II/Creat (Spearman r = 0.41, *p* < 0.0001), and CTX-I and CTX-II/Creat correlated with each other (Spearman r = 0.33, *p* = 0.0001). C2M did not correlate with the other ECM biomarkers. Biomarker levels at baseline were not affected by the concomitant use of antiresorptive treatment (Appendix A).

ALP levels at baseline were higher in older patients and in patients with higher cAKUSSI (Appendix A). ALP levels correlated weakly with joint pain (Spearman r = 0.25, *p* = 0.005) and moderately with osteoarticular disease in the spine (Spearman r = 0.56, *p* < 0.0001), osteopenia of the hips (Spearman r = 0.40, *p* < 0.0001), T-score (Spearman r = −0.43, *p* < 0.0001), and levels of CTX-II/Creat (Spearman r = 0.36, *p* < 0.0001).

## 3. Discussion

This study presents evidence that the remodelling of collagen type I and collagen type II, the main ECM proteins in bone and cartilage, respectively, is altered following nitisinone treatment in patients with alkaptonuria.

The pathogenesis of AKU is characterized by the accumulation of the ochronotic pigment in joints and other cartilaginous tissues. Nitisinone has been approved by the EMA for the treatment of adult patients with AKU, on the basis of the significant lowering of the circulating and urinary HGA, responsibility for the development of ochronosis, and the demonstration of slower disease progression [9]. Nitisinone decreased ochronotic pigment in the eyes and ears, suggesting a modulation of the ochronotic process secondary to HGA lowering. Further, nitisinone also reduced joint and spine pain. It is therefore relevant to examine how the remodelling of the joints is affected by nitisinone. To explore this, non-invasive biomarkers of type I collagen (the main collagen in bone) and type II collagen (the main collagen in cartilage) turnover were measured in the circulation and urine of patients enrolled in the SONIA 2 study.

Type II collagen constitutes up to 90% of the collagen content in cartilage [3], while type I collagen is the main collagen in the bone tissue. Measuring the levels of biomarkers of type I collagen formation (PRO-C1) and resorption (CTX-I) and of type II collagen turnover (CTX-II/Creat and C2M) can therefore illustrate how nitisinone affects the remodelling of cartilage and bone in the joints. It is worth noticing that CTX-II/Creat levels at baseline correlated significantly with the parameters related to both joint and bone health, and hence CTX-II/Creat can be an informative biomarker of the burden of osteoarticular disease. This is in line with the observation that levels of CTX-II/Creat in urine were elevated in patients with more severe disease and in older patients, which is consistent with the increased resorption of calcified cartilage in human joints, observed in ex vivo studies [3] and the observation in AKU mice that joint ochronosis begins in the calcified cartilage [14]. The only biomarker that differed according to sex was CTX-I in serum, which was more elevated in males. This is in line with data from the osteoporosis field where it has been shown that women in general have a lower bone turnover than men until menopause [15,16]. At menopause the overall turnover of bone is increased, which results in net loss of type I collagen measured by CTX-I. In fact, CTX-I is a biomarker used for assessing the effect of bone anti-resorptive treatments such as bisphosphonates [17]. Similar effects are also seen in osteoarthritis; elevated levels of CTX-II/Creat and CTX-I are prognostic for joint space loss [18]. Given the effect of bisphosphonates on CTX-I levels, it was investigated whether concomitant antiresorptive treatment could influence levels and trajectories of CTX-I and the other biomarkers. Levels of the ECM biomarkers and of ALP were not affected by the antiresorptive treatment, and there was no interaction between the concomitant treatment with antiresorptive drugs and the treatment effect of nitisinone on any of the biomarkers.

When looking at trajectories of biomarker change over time, it is possible to observe a sharp increase in the levels of both CTX-I and PRO-C1 after one year of treatment, that then returned to baseline levels at later visits. A similar trend was observed for total levels of the enzyme ALP. The main sources for this enzyme are the liver and bone. There were no signs of any liver damage in nitisinone-treated patients, so a likely explanation for the increase is that it here reflects bone remodelling. Since the three biomarkers presented the same pattern, this phenomenon may reflect an increased remodelling of the bone tissue after starting nitisinone treatment. This may be reflected in the observed increased range of motion of the joints, especially distally, in nitisinone-treated patients, which allowed greater mobility [9].

Type II collagen remodelling was also altered by nitisinone treatment. Especially C2M levels had decreased markedly already at Year 1, suggesting that the treatment influences inflammatory-mediated turnover of cartilage. It was also possible to observe changes in the CTX-II/Creat biomarker that reflect remodelling of articular and calcified cartilage. The difference between C2M and CTX-II is related to their molecular origin. C2M is from the helical domain of type II collagen and is generated by matrix metalloproteinases (MMPs), which are typically upregulated and activated during inflammation, while CTX-II is from the telopeptide end of type II collagen and can be generated by proteases such as cathepsin K expressed by osteoclasts [19,20,21]. In osteoarthritis, CTX-II clusters with bone markers such as PINP and CTX-I more than with cartilage markers [22]. These data indicate that CTX-II mainly originates from the interface between cartilage and bone.

The effect of nitisinone on the biomarker of type II collagen remodelling CTX-II/Creat was more pronounced in younger and male patients, while the reduction of C2M at late time points was more pronounced in patients with lower cAKUSSI at baseline and in male patients. This suggests that the effect of nitisinone on joint health may be more pronounced if the treatment is started earlier when the disease is not yet overt. It is currently not possible to find an explanation for the difference in treatment effect on these biomarkers between males and females.

This study is limited by a relatively low number of examined patients. However, AKU is a rare disease, and this study involves the largest cohorts for which biomarker samples have been collected and the effect of nitisinone has been studied. Moreover, the treatment effect was observed over the course of four years, which may be an insufficient time to see significant changes in remodelling of cartilage and bone tissue, as the drug affects the production of the damaging ochronotic pigment, and it was tested exclusively in patients with chronic pathological signs already present at baseline.

The non-invasive biomarkers shown in this investigation are related to tissue remodelling in joints in patients with AKU. They reflect the turnover of type I collagen, the most abundant collagen in bones, and type II collagen, the most abundant collagen in cartilage, and can describe changes at a molecular level induced by the treatment of nitisinone, likely an indirect effect of blocking the production of the ochronotic pigment. These biomarkers can have clinical utility in the evaluation and management of patients with AKU as they could help find patients in which the disease has already affected the remodelling of joints and who could therefore benefit from an early initiation of nitisinone therapy. Moreover, they can be useful in the evaluation of the long-term effect of nitisinone treatment on joint health, without the need for complex clinical and imaging evaluation.

## 4. Materials and Methods

### 4.1. Clinical Cohort

SONIA 2 (NCT01916382) was a 4-year, open-label, evaluator-blind, multicentre, randomized, no-treatment controlled, parallel-group study to investigate the efficacy and safety of nitisinone in patients with AKU. AKU was diagnosed through a positive urine homogentisic acid (HGA) measurement. During SONIA 2, serum and 24-h urine HGA was measured. Genetic mutations were analysed during the study, but these were not required for diagnosis or entry into SONIA 2. Inclusion and exclusion criteria can be found in [9].

The study was conducted at three sites: Royal Liverpool University Hospital, Liverpool, UK; Hôpital Necker-Enfants Malades, Paris, France; and National Institute of Rheumatic Diseases, Piešťany, Slovakia and aimed to recruit 140 patients with alkaptonuria. In total, 70 patients were to be randomly assigned to receive 10 mg once daily oral nitisinone (Orfadin, Swedish Orphan Biovitrum, Stockholm, Sweden) and 70 to no treatment [9]. Serum (fasting) and urine (first morning void) were collected at baseline and at the yearly visits to the study sites (Year 1, Year 2, Year 3, and Year 4). Missing samples are detailed in Appendix A.

The severity of the disease was established with the composite score cAKUSSI (or the clinical evaluation AKUSSI), which incorporates multiple clinically meaningful AKU symptoms (Appendix A). All items included in the cAKUSSI were assessed at baseline and yearly thereafter.

### 4.2. Patient and Public Involvement

SONIA 2 was performed within the DevelopAKUre consortium, under a Seventh Framework Programme grant by the European Union. Patient societies were part of the consortium and actively involved in patient recruitment and support. All results obtained by the main study and all satellite studies (including the one in this work) are disseminated at annual alkaptonuria meetings, which are attended by AKU patients and researchers.

### 4.3. Measurements

Selected biomarkers of ECM remodelling (Table 6) were measured in serum and urine of SONIA 2 patients (baseline samples and all available samples at later visits) at Nordic Bioscience, Herlev, Denmark. The concentration of CTX-II in urine was normalized for urine creatinine measured using the creatinine Jaffe method on the COBAS 701 analyser at the University of Liverpool, UK. All other markers were measured in serum. All commercial assays were run following the manufacturer’s instructions: β-CrossLaps Elecsys 2010 (Roche Diagnostics, Basel, Switzerland) was used for the measurement of CTX-I, UCartiLaps ELISA (Immunodiagnosticsystem, Tyne & Wear, United Kingdom) for the measurement of CTX-II. PRO-C1 and C2M were analysed by competitive ELISAs developed at Nordic Bioscience (Herlev, Denmark) and performed according to the manufacturers’ protocols.

Alkaline phosphatase (ALP) (total, not bone specific) was measured according to standard methods at each participating clinic as part of the clinical chemistry safety evaluation.

### 4.4. Statistical Analyses

The distribution of the data was analysed, and non-parametric analysis or parametric analysis was performed on log-transformed data.

All relevant study data were tabulated with descriptive statistics, including median and interquartile range (IQR) for the continuous variables, and frequencies and proportions for the categorical variables. Group-wise comparisons were performed using chi-square or Wilcoxon/Mann–Whitney tests. Associations between the biomarkers and clinical variables were assessed using Spearman rank correlation.

Longitudinal linear models (using generalized least squares to account for correlated measurements within patient) were fitted for the analysis of the biomarker data. The analysis was done using log(biomarker) as the dependent variable. Treatment, site, age category (≤55 or >55 years), visit, and treatment-by-visit interaction were added as factors in the model as randomization stratification factors, together with the baseline log(biomarker) value as a covariate to improve the efficiency of the treatment effect estimate. An unstructured covariance matrix was used. Heterogeneity of the treatment effect with respect to sex, age category, or cAKUSSI at baseline was investigated by adding the respective factors and appropriate interaction terms to the base model. Estimates on the log-scale were back-transformed to the original scale to represent geometric means. Contrasts between treatment groups were back-transformed to represent ratios of geometric means.

All statistical analyses were done with R version 4.1.3 (R Core Team (2022). R: A language and environment for statistical computing. R Foundation for Statistical Computing, Vienna, Austria. URL https://www.R-project.org/, (accessed on 1 May 2023)). Two-sided 95% confidence intervals (CIs), corresponding to a nominal two-sided 5% level of significance, were used throughout the analyses. No adjustment for multiple comparisons was made.

## Figures and Tables

**Figure 1 ijms-24-10996-f001:**
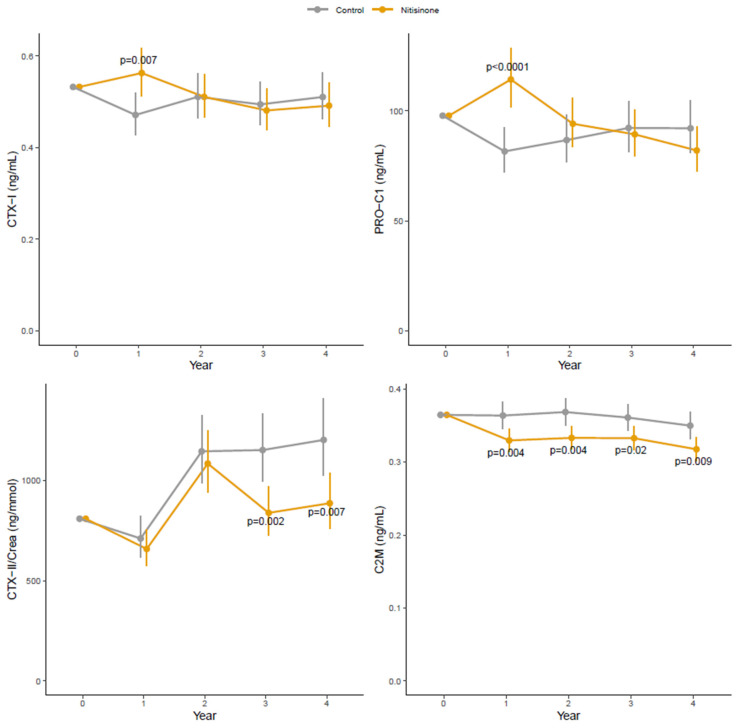
Levels of ECM biomarkers over time in untreated controls and nitisinone-treated patients. Estimated geometric means with 95% confidence intervals. *p*-value for differences between controls and nitisinone-treated patients and the different time points.

**Table 1 ijms-24-10996-t001:** Demographics and clinical characteristics of the patients included in the SONIA 2 biomarker analysis. Statistical significance: * *p* < 0.05.

	Control (N = 69)	Nitisinone (N = 69)	Total (N = 138)
**Sex**			
Male	40 (58.0%)	45 (65.2%)	85 (61.6%)
Female	29 (42.0%)	24 (34.8%)	53 (38.4%)
**Age**			
Median (Q1, Q3)	48.9 (40.7, 56.3)	51.3 (42.0, 58.6)	49.7 (41.2, 56.9)
**Race**			
Asian	2 (2.9%)	1 (1.4%)	3 (2.2%)
Black	0 (0.0%)	1 (1.4%)	1 (0.7%)
White	67 (97.1%)	67 (97.1%)	134 (97.1%)
**Site**			
Liverpool	21 (30.4%)	20 (29.0%)	41 (29.7%)
Paris	16 (23.2%)	16 (23.2%)	32 (23.2%)
Piešťany	32 (46.4%)	33 (47.8%)	65 (47.1%)
**BMI**			
Median (Q1, Q3)	25.9 (24.0, 28.1)	26.8 (24.3, 29.7)	26.2 (24.1, 28.7)
**cAKUSSI at Baseline**			
Median (Q1, Q3)	82.0 (58.8, 107.0)	90.0 (58.0, 108.0)	86.0 (58.0, 107.0)
**cAKUSSI at Year 4**			
Median (Q1, Q3)	101.0 (70.2, 124.8)	93.0 (62.0, 118.0)	97.0 (67.0, 122.0)
**Change in cAKUSSI from baseline**			
Median (Q1, Q3)	15.5 (2.2, 29.5)	7.0 (−4.0, 15.0)	8.0 (−2.5, 24.0) *
**Use of antiresorptive treatment**			
No	39 (56.5%)	38 (55.1%)	77 (55.8%)
Yes	30 (43.5%)	31 (44.9%)	61 (44.2%)

**Table 2 ijms-24-10996-t002:** Concentration of the biomarkers (median and 95% confidence interval (CI)) in the control and nitisinone groups at baseline and at end of study (Year 4). CTX-I, PRO-C1 and C2M (ng/mL), and CTX-II normalized by levels of urine creatinine (CTX-II/Creat) (ng/mmol). Statistically significant differences: ** *p* < 0.01.

Biomarker	Controls		Nitisinone	
	Baseline	End of Study	Baseline	End of Study
**Serum CTX-I**	0.55 [0.43–0.58]	0.52 [0.43–0.56]	0.49 [0.44–0.55]	0.50 [0.43–0.55]
**Serum PRO-C1**	93.1 [77.3–108.0]	81.1 [70.7–97.2]	94.1 [79.0–106.8]	87.4 [66.1–106.1]
**Serum C2M**	0.35 [0.30–0.36]	0.33 [0.27–0.38]	0.35 [0.31–0.39]	0.33 [0.29–0.35]
**Urine CTX-II/Creat**	660.0 [524.8–830.1]	971.9 [756.9–1302.7] **	713.9 [512.2–942.7]	727.4 [565.9–988.8]

**Table 3 ijms-24-10996-t003:** Level of biomarkers at baseline by sex. CTX-I, PRO-C1 and C2M (ng/mL), and CTX-II normalized by levels of urine creatinine (CTX-II/Creat) (ng/mmol). Statistical significance: *** *p* < 0.001.

	Male (N = 85)	Female (N = 53)
**CAKUSSI**		
Median (Q1, Q3)	90.0 (63.8, 112.5)	82.0 (49.0, 106.0)
**Serum CTX_I**		
Median (Q1, Q3)	0.6 (0.4, 0.7)	0.4 (0.3, 0.5) ***
**Serum PRO-C1**		
Median (Q1, Q3)	94.7 (69.4, 123.5)	91.4 (47.9, 126.5)
**Urine CTX-II/Creat**		
Median (Q1, Q3)	748.1 (485.1, 1039.7)	501.9 (370.4, 1159.3)
**Serum C2M**		
Median (Q1, Q3)	0.3 (0.3, 0.4)	0.3 (0.3, 0.4)

**Table 4 ijms-24-10996-t004:** Level of biomarkers at baseline by age. CTX-I, PRO-C1 and C2M (ng/mL), and CTX-II normalized by levels of urine creatinine (CTX-II/Creat) (ng/mmol). Statistical significance: *** *p* < 0.001.

	≤55 (N = 90)	>55 (N = 48)
**CAKUSSI**		
Median (Q1, Q3)	70.0 (49.0, 92.0)	106.0 (88.8, 126.2) ***
**Serum CTX-I**		
Median (Q1, Q3)	0.5 (0.4, 0.6)	0.5 (0.4, 0.6)
**Serum PRO-C1**		
Median (Q1, Q3)	90.9 (55.2, 128.1)	94.5 (77.7, 124.3)
**Urine CTX-II/Creat**		
Median (Q1, Q3)	526.5 (370.6, 834.6)	1059.0 (765.2, 1377.5) ***
**Serum C2M**		
Median (Q1, Q3)	0.3 (0.3, 0.4)	0.4 (0.3, 0.4)

**Table 5 ijms-24-10996-t005:** Level of biomarkers by cAKUSSI at baseline. CTX-I, PRO-C1 and C2M (ng/mL), and CTX-II normalized by levels of urine creatinine (CTX-II/Creat) (ng/mmol). Statistical significance: *** *p* < 0.001.

	Below Median (N = 68)	Above Median (N = 69)
**Serum CTX_I**		
Median (Q1, Q3)	0.5 (0.3, 0.6)	0.5 (0.4, 0.6)
**Serum PRO-C1**		
Median (Q1, Q3)	93.1 (51.8, 130.6)	94.1 (70.1, 124.2)
**Urine CTX-II/Creat**		
Median (Q1, Q3)	440.4 (347.2, 679.8)	944.7 (667.6, 1372.7) ***
**Serum C2M**		
Median (Q1, Q3)	0.3 (0.3, 0.4)	0.3 (0.2, 0.4)

**Table 6 ijms-24-10996-t006:** Biomarkers measured in the SONIA 2 study, description, and normal range (levels in healthy individuals).

Assay	Specifications	Measuring
Serum CTX-I	Cathepsin-generated and cross-linked fragment of type I collagen	Bone resorption
Serum PRO-C1	N-terminal pro-peptide of type I collagen	Bone formation
Urine CTX-II/Creat	C-telopeptide of type II collagen	Cartilage remodelling
Serum C2M	MMP-generated fragment of type II collagen	Cartilage remodelling

## Data Availability

Data are available under request.

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
