# Peer review of "Nitisinone Treatment Affects Biomarkers of Bone and Cartilage Remodelling in Alkaptonuria Patients"

_ijms, 2023, doi:10.3390/ijms241310996_

Round 1
Reviewer 1 Report
Comments
The manuscript presents the follow-up of use of nitisinone on patients of alkaptonuria. The study takes advantage of a drug alleviating the symptoms caused by accumulation of homogentisic acid. The study has the focus on recognizing biomarkers of bone and cartilage to enable early diagnosis of the disease. The goal is highly justified and important for the patient’s treatment.
In Introduction, I would have seen some information on how common is this disease.
Related to the presentation of biomarkers, although described in Table 1, it would be good to mention in all tables whether the samples are from serum or urine. Also in supplemental tables it would be good to mention what are the numbers in brackets (obviously 95% confidential interval).
In Table 1, it would need better separation of the normal range, now it may take some time to understand which values belong to which analyte. Also one may wonder is there a difference in normal range for male and female for PRO-C1 and C2M, and what is the normal range inCTX/Creat in males /the Table gives ranges only for pre- and postmenopausal females).
Figure 2A is misleading, as it is Figure A2 in Supplemental material, correct it.
Basically, I do not like the kind of presentation used in the figures, the Y-axis not starting form zero. Also why is the scale in Figure 1 declining for PRO-C1?
In Discussion the authors interpret that CTX-II/Creat is released due to remodelling of calcified cartilage. How can such an interpretation be done, since I would say that bigger part of type II collagen is present in non-calcified cartilage.
It was mentioned that the effect of nitisinone on. CTX-II/Creat was more pronounced in younger. However, the median age was 50 years, and the authors pinpoint in Introduction the usefulness of non-invasive biomarkers in detecting the early signs of disease. Did the authors try to focus also in the younger patients besides the limit of 55 years? What was the age at the lowest level of the patients?
Minor
Table numbers starts with Table 2, so renumber the tables.
In reference 7, the author Preston AJ has only the initial letter of the family name. Also doi number is wrong, it should be doi: 10.1016/j.joca.2012.04.013
Reference 9 has volume number 5
Line 150, use abbreviation ECM.
Author Response
The manuscript presents the follow-up of use of nitisinone on patients of alkaptonuria. The study takes advantage of a drug alleviating the symptoms caused by accumulation of homogentisic acid. The study has the focus on recognizing biomarkers of bone and cartilage to enable early diagnosis of the disease. The goal is highly justified and important for the patient’s treatment.
In Introduction, I would have seen some information on how common is this disease.
This information is present in Introduction “AKU presents in approximately 1 in every 250,000 to 1 million people worldwide”.
Related to the presentation of biomarkers, although described in Table 1, it would be good to mention in all tables whether the samples are from serum or urine. Also in supplemental tables it would be good to mention what are the numbers in brackets (obviously 95% confidential interval).
Thank you for this comment. We have modified the tables accordingly.
In Table 1, it would need better separation of the normal range, now it may take some time to understand which values belong to which analyte. Also one may wonder is there a difference in normal range for male and female for PRO-C1 and C2M, and what is the normal range inCTX/Creat in males /the Table gives ranges only for pre- and postmenopausal females).
The normal range were provided by the manufacturer. However, since this may generate some confusion, we have decided to remove this information from the table and instead rely on the information mentioned in introduction “Non-invasive biomarkers of extracellular matrix (ECM) remodelling were previously tested in serum and urine of patients enrolled in the 4-week SONIA 1 study[10]. In that study we identified biomarkers of type I and II collagen turnover (CTX-I and PRO-C1, reflecting bone resorption and bone formation, respectively, and CTX-II/Creat and C2M, reflecting cartilage remodelling) that were present at different levels in se-rum and urine of patients with AKU compared to controls[10].”
Figure 2A is misleading, as it is Figure A2 in Supplemental material, correct it.
Thank you for pointing this out, we have now corrected it
Basically, I do not like the kind of presentation used in the figures, the Y-axis not starting form zero. Also why is the scale in Figure 1 declining for PRO-C1?
Thank you for pointing out this mistake, we have now remade Figure 1 starting from 0 and with the correct scale for PRO-C1
In Discussion the authors interpret that CTX-II/Creat is released due to remodelling of calcified cartilage. How can such an interpretation be done, since I would say that bigger part of type II collagen is present in non-calcified cartilage.
To avoid confusion, we decided to change this sentence to “It was also possible to observe changes in the CTX-II/Creat biomarker, which are reflecting remodelling of articular and calcified cartilage. The difference between C2M and CTX-II is related to their molecular origin. C2M is from the helical domain of type II collagen and generated by matrix metalloproteinases (MMPs), which are typically upregulated and activated during inflammation, while CTX-II is from the telopeptide end of type II collagen and can be generated by proteases such cathepsin K expressed by osteoclasts(16–18). In osteoarthritis, CTX-II clustered with bone markers such as PINP and CTX-I more than with cartilage markers (19). These data indicate that CTX-II mainly origins from the interface between cartilage and bone.”
It was mentioned that the effect of nitisinone on. CTX-II/Creat was more pronounced in younger. However, the median age was 50 years, and the authors pinpoint in Introduction the usefulness of non-invasive biomarkers in detecting the early signs of disease. Did the authors try to focus also in the younger patients besides the limit of 55 years? What was the age at the lowest level of the patients?
The cut-off of 55 years was based on the randomization criteria: “Patients were randomly assigned (1:1) to receive nitisinone or no treatment. The randomisation was stratified by study centre and age (≤55 years and >55 years) and was done using randomly permuted blocks (four patients per block) within each study centre and age stratum." We did not pre-specify further subgroup analysis in our statistical analysis plan to look at the younger patients.
Minor
Table numbers starts with Table 2, so renumber the tables.
Done
In reference 7, the author Preston AJ has only the initial letter of the family name. Also doi number is wrong, it should be doi: 10.1016/j.joca.2012.04.013
We changed the input of the reference manager and now the DOI does not appear anymore.
Reference 9 has volume number 5
Done
Line 150, use abbreviation ECM.
Done
Reviewer 2 Report
Manuscript “Nitisinone treatment affects biomarkers of bone and cartilage remodelling in alkaptonuria patients.”
In general the manuscript is difficult to follow and understand.
Specific comments:
Lines 29-33, 39-40: references should be reported.
Lines 58-60: this part should be better explained.
Lines 61-67: SONIA 1 study should be better explained. How many patients were enrolled? It is not reported if there was a difference regarding marker levels in nitisinone vs control group?
Lines 76: this should be table 1 and not 2 etc. The first table presented should be table 1. The authors should give more information enabling the reader to understand the study (considering that the methods are reported at the end of the paper). For example, end of the study is vague.
It is not reported the number of patients enrolled. The demographic/clinical data should be added in the first part of the results.
ALP 1 is not reported in table 2, why?
Figure 1: p-values are unclear. It is unclear if there is a difference between control and nitisone or between different timepoints within the same group.
Line 97: please avoid the use of “we” and use impersonal verbs.
There is confusion in the supplementary files. These tables should be “supplementary table 1” etc. Please start with supplementary table 1 and not 4 etc.
Line 100 :strata is unclear.
Lines 111-114 and table 3 should be moved to the beginning of the results.
Table 3: Statistical analysis (control vs Nitisinone) should be added.
Table 4, 5 and 6: statistical analyses should be added.
Line 144-417: p-value should be reported along with r.
Are there other correlations?
Lines 233-242. How did the authors diagnose AKU? Did the authors measure HGA in urine? Did the authors check genetic mutations? Inclusion/exclusion criteria should be added. It is unclear to me if these patients were treated with nitisinone daily for 4 years.
Control group: How were these patients treated?
Table 1: it is unclear what is the superscript number near each marker.
Section 4.4. ; it is unclear if the authors check normality distribution of the data.
Author Response
In general the manuscript is difficult to follow and understand.
Specific comments:
Lines 29-33, 39-40: references should be reported.
Done
Lines 58-60: this part should be better explained.
We changed the sentence to “Furthermore, since nitisinone is an inhibitor of 4-hydroxyphenylpyruvate dioxygenase, which halts the production of HGA and decreases ochronosis, biomarkers of connec-tive tissue remodelling could describe the downstream effect of the dampening of ochronosis pigment accumulation on the bone and cartilage turnover.”
Lines 61-67: SONIA 1 study should be better explained. How many patients were enrolled? It is not reported if there was a difference regarding marker levels in nitisinone vs control group?
We changed the sentence to “Non-invasive biomarkers of extracellular matrix (ECM) remodelling were previ-ously tested in serum and urine of 40 patients enrolled in the 4-week international, multicentre, randomised, open-label, no-treatment controlled, parallel-group, dose-response SONIA 1 study(13). Among the markers measured in that study, four biomarkers of type I and II collagen turnover (CTX-I and PRO-C1, reflecting bone re-sorption and bone formation, respectively, and CTX-II/Creat and C2M, reflecting cartilage remodelling) were present at different levels in serum and urine of patients with AKU compared to controls(13).”
Lines 76: this should be table 1 and not 2 etc. The first table presented should be table 1. The authors should give more information enabling the reader to understand the study (considering that the methods are reported at the end of the paper). For example, end of the study is vague.
We have changed the table numbering. We have added a subchapter in Results which describes the study and the cohort characteristics.
It is not reported the number of patients enrolled. The demographic/clinical data should be added in the first part of the results.
We have added this to paragraph 2.1
ALP 1 is not reported in table 2, why?
ALP data are only supportive to the discussion and are not part of the paper initial hypothesis, hence we decided to limit the presentation of ALP data to the supplementary material.
Figure 1: p-values are unclear. It is unclear if there is a difference between control and nitisone or between different timepoints within the same group.
We have changed this in the figure caption to make it clearer.
Line 97: please avoid the use of “we” and use impersonal verbs.
We have changed this in the whole manuscript
There is confusion in the supplementary files. These tables should be “supplementary table 1” etc. Please start with supplementary table 1 and not 4 etc.
We changed this to follow the order they appear in the text.
Line 100 :strata is unclear.
We eliminated that sentence.
Lines 111-114 and table 3 should be moved to the beginning of the results.
Done
Table 3: Statistical analysis (control vs Nitisinone) should be added.
Done
Table 4, 5 and 6: statistical analyses should be added.
Done
Line 144-417: p-value should be reported along with r.
Are there other correlations?
Since the correlation matrix contained many parameters, we decided not to report p-value, which would have been overestimated given the large amount of variables, and instead reporting the Spearman r values, which allow for an interpretation of the strenght of the association.
Lines 233-242. How did the authors diagnose AKU? Did the authors measure HGA in urine? Did the authors check genetic mutations? Inclusion/exclusion criteria should be added. It is unclear to me if these patients were treated with nitisinone daily for 4 years.
Control group: How were these patients treated?
We specified this in the beginning of the results section: “138 patients with AKU were enrolled in the 4-year, open-label, evaluator-blind, randomised, no treatment controlled, parallel-group SONIA 2 study. 69 were treated with nitisinone 10 mg daily for 4 years, except 10 patients who developed corneal ker-atopathy and received 2 mg daily instead(9), and 69 were not treated. Since the treat-ment causes a change of colour in the urine, the patients could not be masked, hence the absence of a placebo-controlled group”
Table 1: it is unclear what is the superscript number near each marker.
These were references but are eliminated from the current version of the manuscript
Section 4.4. ; it is unclear if the authors check normality distribution of the data.
Yes, we have added this to the methods
Round 2
Reviewer 2 Report
The authors replied to all my comments.
I have only one pending issue. Lines 296-309: Previously I asked to add p-value along with r. This is fundamental as it is completely unclear if these correlations are significant or not.
Author Response
Thank you for the quick revision.
We have now added the requested p-values in lines 142-159:
"When looking at baseline correlations of the biomarkers of bone and cartilage remodelling with the components of the cAKUSSI related to bone and joint health (osteoarticular disease in joints and spine, pain in joints and spine, number of experienced adult fractures, osteopenia of hips [bone density in g/cm2 and its T-score]) , CTX-I and C2M were not significantly correlated with any of the parameters, PRO-C1 had a weak negative correlation with the bone density T-score (Spearman r= -0.24, p=0.008), while CTX-II/Creat correlated moderately with all parameters except the number of fractures (Spearman r=0.29-0.51, p=0.02-p<0.0001) and had a moderate negative correlation with the bone density T-score (Spearman r= -0.5, p<0.0001). PRO-C1 correlated with both CTX-I (Spearman r=0.72, p<0.0001) and CTX-II/Creat (Spearman r=0.41, p<0.0001), and CTX-I and CTX-II/Creat correlated with each other (Spearman r=0.33, p=0.0001). C2M did not correlate with the other ECM biomarkers. Biomarker levels at baseline were not affected by the concomitant use of antiresorptive treatment (Table A6).
ALP levels at baseline were higher in older patients and in patients with higher cAKUSSI (Table A7). ALP levels correlated weakly with joint pain (Spearman r=0.25, p=0.005) and moderately with osteoarticular disease in spine (Spearman r=0.56, p<0.0001), osteopenia of hips (Spearman r=0.40, p<0.0001) and T-score (Spearman r=-0.43, p<0.0001) and with levels of CTX-II/Creat (Spearman r=0.36, p<0.0001)."